# REPRESENTATION-AGNOSTIC SHAPE FIELDS

**Xiaoyang Huang**[1]**, Jiancheng Yang**[1]**, Yanjun Wang**[1]**, Ziyu Chen**[1]**, Linguo Li**[1]**, Teng Li**[2]**,
Bingbing Ni**[1]*****, Wenjun Zhang**[1]
[1]School of Electronic Information and Electrical Engineering, Shanghai Jiao Tong University
[2]Anhui University
{huangxiaoyang, nibingbing}@sjtu.edu.edu

## ABSTRACT

3D shape analysis has been widely explored in the era of deep learning. Numerous models have been developed for various 3D data representation formats, e.g., MeshCNN for meshes, PointNet for point clouds and VoxNet for voxels. In this study, we present Representation-Agnostic Shape Fields (RASF), a generalizable and computation-efficient shape embedding module for 3D deep learning. RASF is implemented with a learnable 3D grid with multiple channels to store local geometry. Based on RASF, shape embeddings for various 3D shape representations (point clouds, meshes and voxels) are retrieved by coordinate indexing. While there are multiple ways to optimize the learnable parameters of RASF, we provide two effective schemes among all in this paper for RASF pre-training: shape reconstruction and normal estimation. Once trained, RASF becomes a plug-and-play performance booster with negligible cost. Extensive experiments on diverse 3D representation formats, networks and applications, validate the universal effectiveness of the proposed RASF. Code and pre-trained models are publicly available[1].

## 1 INTRODUCTION

3D shape analysis is the foundation to understand the physical world. It has a wide range of applications in real life including robotics (Liu et al., 2020; Liang et al., 2018), autopilot (Shi et al., 2020; Song et al., 2019; Qi et al., 2018; Zhou & Tuzel, 2018), medical imaging (Yang et al., 2021b; Li et al., 2018a; Yang et al., 2021a) and movie animation (Xu et al., 2019; Aberman et al., 2020; Hertz et al., 2020). In recent years, the research on this topic has prevailed and showed promising results in various tasks, such as object classification, part segmentation, scene segmentation, etc.

Shape could be represented in different data formats, among which meshes, point clouds and voxels are most commonly used. In the deep learning era, most studies on these three representations use coordinates or coordinates-like feature as input to feed into the backbone network. For point clouds, the direct input to the network is the point coordinates. For meshes, the node feature of the graph is the vertices coordinates. For volumetric data, the 3D shape is denoted by whether a voxel in a particular position is occupied or not. Using coordinates to characterize a shape is simple and straightforward. However, the major problem with this is that coordinate lacks contextual geometric information. Hence the capacity of the backbone network could be restricted. Even though various operators and

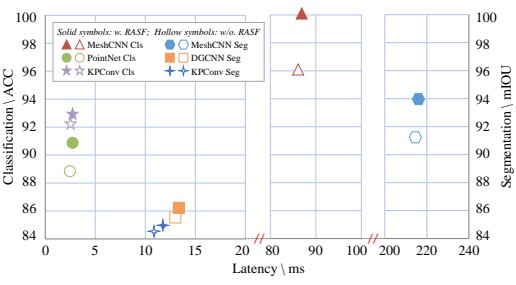

Figure 1: Vertical Axis: Performance; Horizontal Axis: Latency. Diverse backbones with (solid symbols) and without RASF (hollow symbols) are evaluated. RASF could be seamlessly plugged into any 3D deep learning pipeline to improve performance across diverse downstream tasks and datasets, with little code modification and computation cost.

---

*Corresponding Author
[1]https://github.com/seanywang0408/RASF

backbones are proposed to extract high-level feature from the combination of coordinates by aggregating the local geometry, the effect of the input feature is not clear yet.

The practice in Natural Language Processing (NLP) and Data Mining (DM) might shed some light on this issue. Word embeddings (Mikolov et al., 2013; Pennington et al., 2014) is proposed very early in NLP to map words into an embedding space using embedding layers, which work as lookup tables that are indexed by the one-hot encoding of words. The word embeddings retrieved from embedding layers have similar values for words with similar meanings. This technique is widely adopted across the NLP area and notably boosts the overall performance, regardless of how the text data is distributed and which language model is used (Brown et al., 2020; Devlin et al., 2019; Radford et al., 2019; Vaswani et al., 2017). Studies in the field of data mining (DM) also learn continuous embedding representation for graph nodes (Grover & Leskovec, 2016; Perozzi et al., 2014). Embedding learning in NLP and DM indicates that **the effect of input feature and the capacity of the backbone network are somehow orthogonal to each other**. It raises the question that whether there is a better way to denote a shape instead of vanilla coordinates, so as to facilitate the backbone network to execute downstream tasks, regardless of which backbone model is used.

In this work, we introduce Representation-Agnostic Shape Fields (RASF), a shape embedding layer that maps coordinates to shape embeddings with rich geometry information. RASF is implemented using a learnable multi-channel 3D grid. Similar to the lookup table in word embedding layer, coordinates within a local shape index from this 3D grid and retrieve shape embeddings. With simple operation on data, we make RASF compatible with major 3D representations, including point clouds, meshes and voxels. To obtain the weights of RASF, we investigate several pre-training schemes for RASF, among which we find that self-supervised training schemes, *i.e.*, reconstruction and normal estimation, yield the best performance and generalizability. Once trained, RASF could be seamlessly plugged into any 3D deep learning pipeline to improve performance across diverse downstream tasks and datasets, with little code modification and computation cost (See Fig. 1). We empirically show that RASF consistently brings significant improvement under diverse backbones and applications, including object classification, part segmentation and scene segmentation.

**Contributions**   In this work, we introduce a *generalizable* (i.e., it could be used in different 3D representations, backbones and downstream tasks) and *computation-efficient* shape embedding layer for 3D deep learning, named Representation-Agnostic Shape Fields (RASF). It applies a learnable multi-channel 3D grid to store local geometry. Shape embeddings for various 3D shape representations (point clouds, meshes and voxels) are retrieved by coordinates indexing. While there are multiple ways to obtain the coefficients of RASF, we provide two effective schemes among all in this paper for RASF pre-training, that is shape reconstruction and normal estimation. Abundant experiments across different representations, backbones and downstream tasks are conducted to validate the generalization and efficiency of our proposed RASF.

## 2   RELATED WORK

### 2.1   3D SHAPE ANALYSIS

**Point clouds**   Point cloud data could be directly obtained from 3D LiDAR sensors. Due to its compactness that represents only the surface of the objects and its aligned format (an $N \times 3$ matrix) that suits the common deep learning frameworks, point clouds are the most extensively discussed representation in the field of 3D shape analysis. PointNet (Qi et al., 2017a) is the pioneering deep learning network to process point clouds. It learns the feature of each point with a shared MLP and aggregates all points by global max-pooling. PointNet++ (Qi et al., 2017b) supports hierarchical points aggregation to extract geometric information at different scales. DGCNN (Wang et al., 2019) builds a dynamic graph in each layer of the network to incorporate neighbor nodes by its proposed EdgeConv module. PointCNN (Li et al., 2018b), RSCNN (Liu et al., 2019c), DPAM (Liu et al., 2019a), ShellNet (Zhang et al., 2019) and KPConv (Thomas et al., 2019) extend 3D convolution operation to the irregular point clouds data in different ways, while PAT (Yang et al., 2019) and PCT (Guo et al., 2021) leverage transformer to process point clouds. Shape Self-Correction (Chen et al., 2021) proposed a self-supervised method to for point clouds analysis.

**Meshes**   Meshes are mostly used in computer graphics (Kato et al., 2018; Liu et al., 2019b; Pfaff et al., 2020; Smirnov & Solomon, 2021). GWCNN (Ezuz et al., 2017) maps unstructured geometric data to a regular domain for nonrigid shape analysis. MeshCNN (Hanocka et al., 2019) adopts specialized convolution and pooling operations on mesh edges. The convolution is conducted on the

edge and the four adjacent edges around the incident triangles, while the pooling operation generates new geometry via adaptive edge collapse. Besides, MeshCNN designs several hand-crafted features that characterize the edge geometry, that is the dihedral angle, two inner angles and two edge-length ratios for each face. The 5-dimensional vector is fed into the MeshCNN network as input. Alternatively, MeshNet (Feng et al., 2019) regards faces as units. It introduces face-unit and feature-splitting to learn on meshes directly. A more recent work, HodgeNet (Smirnov & Solomon, 2021), operates on feature of vertices and edges simultaneously. There are also works in the field of single image reconstruction, which consider mesh data as graphs (Wang et al., 2018; Gkioxari et al., 2019; Pan et al., 2019; Wen et al., 2019). In this case, graph convolutions on nodes (vertexes) are adopted to aggregate local geometry.

**Voxels** Volumetric data provides regular grids to represent 3D shapes. They could be processed by methods analogous to 2D grids. 3DShapeNets (Wu et al., 2015) proposed to represent 3D shapes with volumetric grids and introduced 3D Convolution Neural Networks (3DCNN) for voxel classification. VoxNet (Maturana & Scherer, 2015) utilized 3DCNN for robust 3D object recognition. The Voxception-ResNet (VRN) (Brock et al., 2016) introduced popular 2D network blocks into volumetric networks. The drawback of volumetric representations lies in that 3DCNN is computationally expensive. In recent years, LP-3DCNN (Kumawat & Raman, 2019) is proposed to alleviate the computation issue. It applies 3D Short Term Fourier Transform (STFT) to replace the 3D convolution layers.

## 2.2 SHAPE DESCRIPTORS AND EMBEDDING LEARNING

Surface feature descriptors or shape descriptors for non-rigid objects (generally in mesh format) are another line of work in 3D shape analysis, often applied in shape correspondence, retrieval and segmentation. Shape descriptors aim at representing the local geometry around a point in a non-rigid shape. Classic surface descriptors describe a local shape patch based on diffusion and spectral geometry to achieve isometry-invariance, for example, heat kernel signature (HKS) (Sun et al., 2009), wave kernel signature (WKS) (Aubry et al., 2011) and optimal spectral descriptors (OSD) (Litman & Bronstein, 2013). Masci et al (Masci et al., 2015) proposed to automatically learn various shape descriptors for Riemannian manifolds using a geodesic convolutional neural network.

Studies in NLP embedding learning maps the one-hot encoding of words to a real-value vector via a learnable embedding layer. An embedding layer is a lookup table, using words as indices and retrieve word embeddings from it. The embedding layer could be trained in an unsupervised manner for general proposes, or in a supervised manner for a specific task, e.g., document classification.

## 3 METHODOLOGY

### 3.1 REPRESENTATION-AGNOSTIC SHAPE FIELDS (RASF)

We propose a shape embedding layer named Representation-Agnostic Shape Fields (RASF), to facilitate 3D shape analysis across various representations. As shown in Fig. 2 (a), RASF is implemented as a trainable 3D grid with the shape of $R \times R \times R \times C$, where $R$ is the resolution of the grids and C is the channel dimension. Common 3D shape representations are based on point coordinates. Suppose the shape is denoted as a point set $X \in \mathbb{R}^{N \times 3}$. For each point $\mathcal{P}$ in the shape, we extract the K-Nearest-Neighbor points $\mathcal{P}_{neigh}$ from the whole point set. Regarding $\mathcal{P}$ as the central point, we normalize the coordinates of the local point set $\mathcal{A} = [\mathcal{P}, \mathcal{P}_{neigh}]$ to the range of the grid. In other words, $\mathcal{P}$ would be placed at the center of the grid, while the farthest neighbor to $\mathcal{P}$ would be placed at the border of the grid. Similar to the word embedding layer that is indexed by one-hot encoding of words, the normalized local point set serves as indexes to retrieve shape features from the grid. The difference is that one-hot encoding is discrete while coordinates are continuous. Therefore the indexing is accomplished via trilinear interpolation. This operation inspired by Spatial Transformer Networks (STNs) (Jaderberg et al., 2015) could be efficiently implemented by modern deep learning frameworks (*e.g.*, *grid_sample* in PyTorch (Paszke et al., 2017)). This function enables continuous indexing in batches to retrieve feature from discrete regular grid data. The trilinear interpolation returns a $K \times C$ matrix for $\mathcal{P}$ ($K$ includes the point $\mathcal{P}$ itself), which is then reduced to $1 \times C$ by *max-pooling* on the first dimension. The $C - dim$ vector, named shape embedding (SE), encodes the shape in the local area $\mathcal{A}$ around the central point $\mathcal{P}$. The whole process to obtain shape

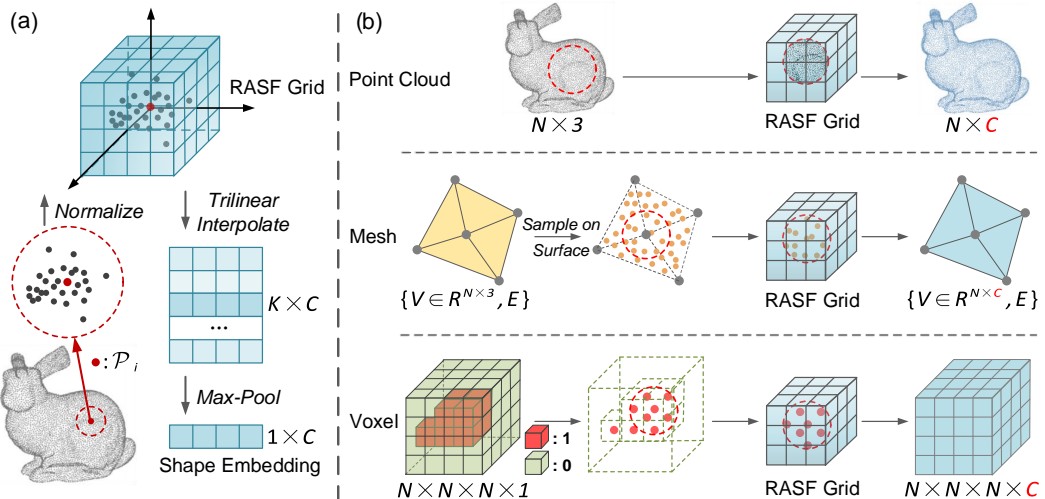

Figure 2: **(a) Inference of RASF.** The process to obtain shape embedding of $\mathcal{P}$ via RASF grid. We extract the K-Nearest-Neighbor of $\mathcal{P}$ from the whole point set and normalize the local point set around $\mathcal{P}$ to the range of RASF grid. In other words, $\mathcal{P}$ would be placed at the center of the grid, while the farthest neighbor to $\mathcal{P}$ would be placed at the border of the grid. The normalized local point set serves as indexes to fetch feature from the grid via trilinear interpolation. The $K \times C$ interpolated feature is reduced to $1 \times C$ by *max-pooling*, named shape embedding. **(b)** RASF implementation for point clouds (top), meshes (middle) and voxels (bottom). $C$ denotes the channel of RASF. **Point Clouds**: $N$ denotes the number of points. **Meshes**: $V, E$ denotes the vertexes and edges of meshes. We re-sample points on the faces of meshes and combine them with vertexes to fetch vertexes feature from RASF. **Voxels**: $N$ denotes the size of voxels. We leverage the points with value of 1 to fetch voxel feature from RASF.

embedding $SE(\mathcal{P})$ of point $\mathcal{P}$ could be formulated as follows:

$$SE(\mathcal{P}) = max(\mathcal{GS}(\mathcal{N}([\mathcal{P}, \mathcal{P}_{neigh}]))), \tag{1}$$

where $\mathcal{GS}$ denotes the *grid_sample* function and $\mathcal{N}$ denotes the normalization. Note that the central point always corresponds to the central feature of RASF due to the normalization. Since RASF is a cubic grid, we use L1-distance in KNN searching for parameters efficiency. After processing each point in this way, we obtain shape embeddings $SE \in \mathbb{R}^{N \times C}$ for all the points. The computation in RASF is negligible compared to the backbone networks, as analyzed in Sec. 5.

## 3.2 RASF FOR VARIOUS REPRESENTATIONS

**RASF for Point Clouds.**     3D shape using point clouds representation could be denoted by a $N \times 3$ matrix, where $N$ is the number of points. Therefore applying RASF on point clouds is natural. During the downstream tasks, the $N \times C$ matrix is fed into the backbone network together with coordinates as auxiliary features.

**RASF for Meshes.**    Suppose a mesh $\{V, E, F\}$ is denoted by three elements: vertexes $V \in \mathbb{R}^{N_V \times 3}$, edges $E$ and faces $F$. Since mesh-based backbone networks receive different elements as units, there are minor adjustments on how the shape embedding is obtained in the RASF. Some backbone networks accept vertex features as input. Since the vertex positions in meshes are irregular (meaning that some vertexes are closer to others, while some are further), it is hard to extract meaningful shapes given only the vertex positions. In this regard, we re-sample denser points $P \in \mathbb{R}^{N_P \times 3}$ on the faces of meshes and combine the re-sampled points along with the vertexes together, as shown in the middle of Fig. 2 (b). Then we feed the combined point set $[V, P]$ into RASF, to obtain shape embedding of the vertexes $SE \in \mathbb{R}^{N_V \times C}$. Some backbone networks accept edge features as input, e.g. MeshCNN (Hanocka et al., 2019). For similar reasons, we combine the midpoint of the edges with the re-sampled points $P$ to obtain the shape embedding of the edges in a similar way. For those backbone networks that receive face feature as input (Feng et al., 2019),

we combine the barycenter of faces with the re-sampled points $P$ to obtain the shape embedding of faces. Note that the shape embeddings obtained through the above approaches are entirely compatible with any mesh-based backbone networks, owing to the flexibility of RASF.

**RASF for Voxels.** An equivalent representation to volumetric data is that points lie in positions where the voxel value is 1 while no point exists when the voxel value is 0. We leverage these "virtual" points to fetch voxel features from RASF during the downstream tasks, as shown at the bottom of Fig. 2 (b). Besides, we adjust the receptive field of RASF to a determinant distance instead of K-Nearest-Neighbors. In this case, voxels that are outside and far from the shape surface (no occupied voxel exists within the receptive field of RASF), would yield a shape embedding of a zero vector. Voxels that are inside and far from the shape surface (all voxels around it are occupied), would yield a shape embedding of the same value. The $N \times N \times N \times 1$ volumetric data is transformed to an $N \times N \times N \times C$ tensor and fed into the backbone network.

### 3.3 LEARNING RASF IN PRETEXT TASKS

**Reconstruction.** We provide several schemes to pre-train RASF. The major scheme that we use in most of the experiments is reconstruction. Given the shape embeddings of each point $SE \in \mathbb{R}^{N \times C}$, we randomly sample $N_s$ embeddings from $N$, and concatenate the embeddings with the coordinates. The coordinates are indispensable since the shape embeddings only encode the local geometry, without knowledge of the global geometry. The $N_s \times (C + 3)$ tensor is fed into the reconstruction network, which consists of a shared MLP, a max-pooling layer, and several linear layers. The last linear layer outputs $(N * 3)$ elements. We reshaped the output to $N \times 3$, which is the predicted point set $X_{pred}$. The loss function is the chamfer distance between $X_{pred}$ and the ground-truth $X$.

**Normal Estimation.** This pretext task is to predict the normal of each point. We feed the shape embeddings and coordinates of each point $SE \in \mathbb{R}^{N \times (C+3)}$ into a shared MLP, which outputs the predicted normals $\mathcal{N}_{pred} \in \mathbb{R}^{N \times 3}$. We use cosine similarity as the loss function.

**Supervised.** The supervised task is shape classification, given only a small portion of points and their shape embeddings. The overall process is quite similar to that of the reconstruction task, only that the output of the network is the probability of each class, instead of coordinates.

Each of these pre-training pushes RASF to encode the local geometry. We empirically find that self-supervised pre-training schemes (reconstruction and normal estimation) outperform the supervised one (classification). The self-supervised training enables RASF to be robust and transferable across different datasets and downstream tasks. For simplicity, we show the experimental results of reconstruction in Sec. 4 and then analyze different ways of pre-training in ablation study (Sec. 5). In practical implementations, we pre-train RASF using ShapeNetPart (Yi et al., 2016) and fix its weights in downstream tasks. (All the downstream tasks use the same RASF.) Empirical study shows that fine-tuning the weights during the downstream tasks leads to unstable performance. The detailed pre-training settings and hyper-parameter analysis are presented in the appendix A.1, A.6.

## 4 EXPERIMENTS

### 4.1 PERFORMANCE OF PRETEXT TASKS

We analyze the performance of pretext tasks by visualizing RASF weights in two ways (Fig. 3) and verify RASF using a linear evaluation protocol.

The first visualization is to directly illustrate the channels in 3D and 2D. The 3D illustrations use *visvis* (Klein, 2020) to show the three-dimensional grids, while the 2D illustrations show the max-pooling output on one axis of the grids. We figure that each channel focuses on a different area, representing a particular local geometry. The other visualization is to investigate RASF's response to geometrically-varying shapes. Practically, we feed RASF with a set of semi-ellipsoids with different curvatures, as shown in Fig. 3. It is observed that some of the channels (channel 6, 11, 12, 20, 25, 29, 30) have strong correspondence to the ellipsoid curvature, *i.e.*, the response of these channels changes gradually from large curvature to small curvature. On the other hand, some channels have the same response given different curvatures. We conjecture these channels could be related to other geometries, such as cones or cubes. (We randomly choose some channels for demonstrations. Visualizations of all channels and detailed settings could be found in appendix A.4.)

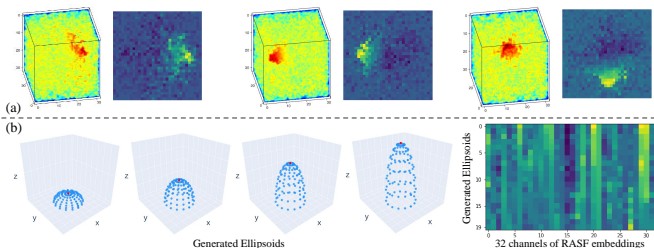

Figure 3: (a) RASF visualization in 3D and 2D. (b) We generate multiple Ellipsoids by deformation along Z-Axis and see their responses in RASF embeddings. Some channels have strong correspondence to the ellipsoid curvature.

Table 1: Linear classification protocol. We test several kinds of linear classifier and report the ACC on ModelNet40. It is shown that RASF feature is more discriminative than raw feature.

|  | Classifier | Raw | +RASF |
|---|---|---|---|
| PLY | Max + FC | 25.57 | 32.17 |
|  | Point-Wise FC + Max + FC | 85.90 | 87.28 |
|  | Flattened Feature & FC | 9.52 | 39.10 |
| Voxel | Flattened Feature & FC | 67.65 | 68.33 |

Moreover, we verify RASF by a linear classification protocol on point clouds and voxels data. For point clouds, we experiment on three kinds of linear classifier: 1) max-pooling on the number of points followed by a fully-connected (FC) layer; 2) a point-wise (shared) FC layer followed by max-pooling and a FC layer; 3) flatten the $N \times C$ input to one dimension and use a FC layer as classifier. For voxels data, we simply flatten the $N \times N \times N \times C$ input to one dimension and use a FC layer as a classifier. The results are shown in Table 1. It is observed that RASF feature outperforms raw feature in all settings, indicating RASF feature is more discriminative than raw feature. Especially for flattened points cloud, the out-of-order raw feature leads to extreme performance degradation, while RASF feature still yields an accuracy of 39.10.

## 4.2 DATASETS IN DOWNSTREAM TASKS

We use five different datasets to evaluate the general effectiveness of RASF, which are various on tasks, characteristics, and shape representations. The summarized introduction is listed in Table 2. All settings are identical for each backbone with and without RASF, including training hyper-parameters, train-test splits, and so on.

Table 2: Datasets used in this study, with their train-test splits, data representation formats, descriptions and tasks. See Appendix A.5 for detailed demonstrations of each dataset.

| Dataset | Train | Test | Representation | Description | Task | Metric |
|---|---|---|---|---|---|---|
| ModelNet10 (Wu et al., 2015) | 3,991 | 908 | Point Cloud & Voxel | 3D Objects (Rigid) | Classification | ACC |
| ModelNet40 (Wu et al., 2015) | 9,843 | 2,468 | Point Cloud & Voxel | 3D Objects (Rigid) | Classification | ACC |
| ShapeNetPart (Yi et al., 2016) | 12,137 | 2,874 | Point Cloud | 3D Objects (Rigid) | Part Segmentation | mIOU |
| S3DIS (Armeni et al., 2016) | 6-Fold Cross-Val | | Point Cloud | Indoor Scene (Rigid) | Semantic Segmentation | mIOU |
| SHREC10 (Lian et al., 2011) | 300 | 300 | Mesh | 3D Objects (Rigid & Non-Rigid) | Classification | ACC |
| SHREC16 (Lian et al., 2011) | 480 | 120 | Mesh | 3D Objects (Rigid & Non-Rigid) | Classification | ACC |
| HUMAN (Maron et al., 2017) | 370 | 18 | Mesh | Human Bodies (Non-Rigid) | Part Segmentation | mIOU |

## 4.3 TRANSFER TO POINT CLOUDS

**Settings.** For point cloud representation, we conduct experiments on ModelNet40 for classification, ShapeNetPart for part segmentation and S3DIS for semantic scene segmentation. We compare the performance with and without the RASF input on various point cloud backbones. For classification, we experiment on PointNet, PointNet++, KPConv, DGCNN and PCT. For part segmentation, we experiment on PointNet, PointNet++, KPConv and DGCNN. For scene segmentation, we evaluate the performance of RASF in DGCNN. All the results are obtained by running the public official code ourselves. For RASF, we simply increase the input channel of all the backbone by $C$ (the channel dimension of RASF), while other hyper-parameters remain unchanged and default. Normals are excluded for all the backbones. The number of sample points and the data augmentation techniques for each backbone are aligned with its original setting.

**Results.** As shown in Table 3 and Table 4. RASF consistently improves the performance of all the backbone models on different tasks. For PointNet which lacks local operations, RASF notably improve 2.14% over baseline. For other backbones which contain various neighbor-based operators, RASF still brings consistent improvement, even though RASF is also based on local operations. It demonstrates that the effect of input feature and the capacity of the backbone network could

Table 3: Results on point clouds data of 3D objects. We compare performance on different point-clouds-based backbones and downstream tasks with and without RASF.

| Method | ModelNet40 ACC | ShapeNetPart mIOU |
|---|---|---|
| PointNet | 88.78 | 84.19 |
| PointNet +RASF | 90.92 | 84.85 |
| Gain | **+2.14** | **+0.66** |
| PointNet++ | 91.55 | 84.86 |
| PointNet++ +RASF | 91.82 | 85.32 |
| Gain | **+0.27** | **+0.46** |
| KPConv | 92.18 | 85.57 |
| KPConv +RASF | 92.80 | 86.24 |
| Gain | **+0.62** | **+0.67** |
| DGCNN | 92.71 | 84.50 |
| DGCNN +RASF | 92.95 | 84.91 |
| Gain | **+0.24** | **+0.41** |
| PCT | 92.93 | - |
| PCT +RASF | 93.18 | - |
| Gain | **+0.25** | - |

Table 4: Results on point clouds data of 3D scenes (S3DIS). We compare performance on DGCNN with and without RASF.

| Method | mIOU | Overall ACC |
|---|---|---|
| DGCNN | 59.04 | 86.71 |
| DGCNN +RASF | 59.73 | 86.82 |
| Gain | **+0.69** | **+0.11** |

Table 5: Results on meshes data with MeshCNN.

| Method | SHREC10 ACC | SHREC16 ACC | HUMAN mIOU |
|---|---|---|---|
| MeshCNN | 96.00 | 99.17 | 91.35 |
| MeshCNN +RASF | 100.00 | 100.00 | 93.90 |
| Gain | **+4.00** | **+0.83** | **+2.55** |

Table 6: Results on voxels with VoxNet.

| Method | ModelNet10 | ModelNet40 |
|---|---|---|
| VoxNet | 91.48 | 82.78 |
| VoxNet +RASF | 92.46 | 83.07 |
| Gain | **+0.98** | **+0.29** |

complement each other. In the scene segmentation task, where the point clouds contain richer and more complex local geometry, RASF notably improves the result over DGCNN. It demonstrates that RASF remains robust even when the downstream data distribution is thoroughly distinct from the pre-training data. With the additional shape embedding input, networks have a better understanding of the shape semantics than just using coordinates. RASF is able to generally boost the performance under various backbones and tasks with little memory and computation cost.

## 4.4 Transfer to Meshes

**Settings.** For Mesh Representation, we use MeshCNN to evaluate the effectiveness of RASF. We conduct classification on SHREC dataset and segmentation on HUMAN dataset, following the settings in MeshCNN. MeshCNN considers the edge as the unit of meshes. It manually calculates the edge feature based on the two adjacent triangles. For classification, we train the model for 200 epochs using Adam (Kingma & Ba, 2014) optimizer with an initial learning rate of $0.002$ and linearly decrease the learning rate to 0 from the $100th$ epoch. For segmentation, the number of epochs is 600 while the initial learning rate is set to $0.002$. For RASF, we extract shape embeddings of the edges following our proposed method for mesh, concatenate it with its original feature, then increase the channel of the first layer of the model to feed them in the network.

**Results.** As shown in Table 5, RASF notably boosts the performance of MeshCNN over diverse datasets and tasks. Especially, MeshCNN with RASF reaches a classification accuracy of $100\%$ on SHREC10 and SHREC16. Besides, we observed that the increment of the performance on meshes is much more significant compared to other representations. For one reason, the surface sampling of meshes yields a similar data distribution to point clouds, meaning little mismatch between pre-training and downstream training in terms of shape embedding. For another, the sampling on the mesh surface actually introduces new geometry compared to using vertexes and edges only. Through RASF, the geometry is encoded in the shape embedding, resulting in a higher performance boost. Moreover, RASF could be integrated to mesh networks seamlessly without sacrificing efficiency in terms of training time and model size.

Noticed that MeshCNN accepts not only coordinates input but also the hand-crafted edge feature. In this case, RASF is still able to increase the performance of the network. The comparison between the hand-crafted feature and RASF will be thoroughly discussed in Sec. 5.

## 4.5 TRANSFER TO VOXELS

**Settings.** We adopt VoxNet to evaluate RASF on voxels data, using ModelNet10 and ModelNet40 for classification task. The voxels data is in a shape of $32 \times 32 \times 32$. We train the model for $100$ epochs using Adam optimizer with an initial learning rate of 0.001 and multiply the learning rate by 0.5 for every 20 epochs. We obtain shape embedding of each voxel from RASF following the proposed RASF implementation for voxels. Then we feed the $32 \times 32 \times 32 \times 32$ shape embedding into VoxNet, by changing the input channel of the first layer. The best accuracy on test-set among all epochs is reported.

**Results.** RASF boosts the performance of VoxNet on both datasets (Table 6). Note that RASF is pre-trained on point clouds, in which the scale is different from voxels. What's more, point clouds only exist on the surface of the objects, while voxels are solid, which indicates they have entirely different distributions. Even in this case, RASF could still increase the performance by aggregating local geometry. It enriches the shape semantics of voxel representation by transferring the learned geometry to distinct data distributions, demonstrating the robustness of RASF.

## 5 DISCUSSION

### 5.1 ABLATION STUDY

**Module Design** The adoption of a learnable grid in the shape embedding layer is motivated from embedding layer in NLP. In this part we investigate how a learnable grid is superior to other module architecture choices, including a PointNet model (consisting of a point-wise MLP, a Max-pooling layer and global MLP) and a simplified PointNet-like module (consisting of a single fully-connected layer and a Max-pooling layer). Besides, we include EdgeConv module in DGCNN, which has been proved to be an effective neighborhood-based point clouds operator. We pre-train these three modules using the same reconstruction pre-text task described in Sec. 3. As the results shown in Table 7, RASF has better performance in each downstream task. EdgeConv yields comparable performance on point clouds, yet it rapidly deteriorates on mesh data. We argue that RASF is a better choice with respect to generalizability.

**Comparison of Pre-Training Schemes** We compare the performance of the three pre-training schemes, including reconstruction, normal estimation and classification. We experiment on ModelNet40 for point clouds, SHREC10 for meshes and ModelNet10 for voxels. We use PointNet, MeshCNN and VoxNet as the backbones for point clouds, meshes and voxels respectively. As shown in the middle bar of Table 8, all the pre-training schemes improve the performance over baseline. RASF obtained from self-supervised pre-training consistently outperforms the supervised one. We argue that self-supervised pre-training enables RASF to learn more general and robust embeddings that are more related to the local geometry while less related to high-level semantics.

**Fixed weights vs. Fine-Tuned vs. Random-Initialized.** We evaluate the effect of the pre-trained RASF weights by comparing fixed, fine-tuned, and random-initialized RASF in the downstream tasks. Fixed RASF is the setting we use in all the experiments. Fine-tuned RASF refers to optimizing the pre-trained RASF together with the backbone network in the training of the downstream tasks, while random-initialized refers to a non-pre-trained RASF that is optimized within the downstream tasks. The detailed settings are the same as described in the previous paragraph. As shown in the bottom bar of Table 8, random-initialized RASFs are consistently worse than fixed RASFs by a large margin. It proves that RASF considered as a feature layer would not work without pre-training. Besides, fine-tuned RASFs yield unstable results compared to fixed RASFs. This phenomenon demonstrates the pre-trained RASF generalizes well in downstream tasks.

### 5.2 COMPLEXITY ANALYSIS

We analyze the complexity of RASF. The most computationally-intense step in RASF occurs in K-Nearest-Neighbors algorithm, in which the time complexity is $\mathcal{O}(N^2)$, while the trilinear step yields $\mathcal{O}(8NK)$. The number of parameters in RASF is $R \times R \times R \times C$. As RASF could be fixed in the downstream tasks, it introduces no additional memory cost apart from its parameters. We evaluate the actual running time of RASF and multiple backbones with a batch size of $1$ (Table 9). Note that our current implementation of K-Nearest-Neighbors is a naive version with PyTorch. The

Table 7: Analysis of module design as shape embedding layers.

| Methods | Point Cloud ModelNet40 | Mesh SHREC10 | Mesh HUMAN |
|---|---|---|---|
| Baseline | 88.78 | 96.00 | 91.35 |
| PointNet | 87.84 | - | - |
| FC + Max | 90.59 | 98.00 | 93.68 |
| EdgeConv | 90.90 | 85.33 | 80.53 |
| RASF | **90.92** | **100.00** | **93.90** |

Table 8: Analysis of RASF pre-training schemes (top) and the effects of fixed / fine-tuned / random-initialized weights (bottom).

| Method | Point Cloud ModelNet40 | Mesh SHREC10 | Voxel ModelNet10 |
|---|---|---|---|
| Baseline | 88.78 | 96.00 | 91.48 |
| Reconstruction | 90.92 | **100.00** | **92.46** |
| Normal Estim. | **91.97** | 99.67 | 92.32 |
| Supervised | 89.34 | 98.00 | 92.21 |
| Fixed | **90.28** | 98.17 | **92.46** |
| Fine-Tuned | 89.64 | **99.33** | 91.36 |
| Random | 88.47 | 95.67 | 92.09 |

Table 9: Time complexity analysis of RASF. The actual running time of RASF is almost negligible compared to the backbones.

| Task | Latency *of* | Backbone | Backbone+RASF |
|---|---|---|---|
| Cls. | PointNet | 2.47ms | 2.75ms (+11.3%) |
| | KPConv | 2.54ms | 2.71ms (+6.6%) |
| | DGCNN | 2.96ms | 3.43ms (+15.8%) |
| | MeshCNN | 86.14ms | 86.98ms (+0.9%) |
| Seg. | PointNet | 2.54ms | 2.80ms +10.2%) |
| | KPConv | 12.8ms | 13.5ms (+5.4%) |
| | DGCNN | 10.9ms | 11.8ms (+8.2%) |
| | MeshCNN | 214ms | 217ms (+1.4%) |

Table 10: Comparison between RASF and hand-crafted (HC) features. The baselines are PointNet for ModelNet40 (ACC), PointNet for ShapeNetPart (mIOU) and MeshCNN for SHREC10 (ACC).

| Method | Point Cloud ModelNet40 | Point Cloud ShapeNetPart | Mesh SHREC10 |
|---|---|---|---|
| Baseline | 88.78 | 84.19 | 64.67 |
| +HC | 89.85 | 84.66 | 96.00 |
| +RASF | 90.92 | 84.85 | 97.33 |
| +HC +RASF | **91.33** | **85.00** | **100.00** |

actual time cost is expected to be lower by integrating more sophisticated KNN implementation, e.g., *Faiss* (Johnson et al., 2017). All the results are measured on an RTX 2080 Ti.

## 5.3 COMPARISON WITH HAND-CRAFTED FEATURES

In point clouds analysis, the point normals are widely used as auxiliary features to improve performance. In meshes analysis, MeshCNN proposed to use the geometric statistics around the edges as auxiliary features, that is the dihedral angle, two inner angles and two edge-length ratios for each face. In this work, we learn local geometry by optimizing the RASF grid in pre-text tasks. We compare shape embeddings from RASF with these two hand-crafted features. For the point cloud baseline, we only feed the point coordinates as input, using a PointNet backbone. For the mesh baseline, we remove the hand-crafted features on edges and replace them with the mid-points of the edges, representing the position of the edges. We experiment on SHREC10 for meshes and ModelNet40 (classification), ShapeNetPart (segmentation) for point clouds.

Backbones with RASF consistently outperform baselines by large margins (Table 10). Hand-crafted features also improve the performance, but not as good as RASF, indicating that RASF provides richer geometric information than hand-crafted features. We reckon that RASF is pre-trained through a reconstruction task which requires RASF to provide comprehensive local geometric information. We also noticed that combining RASF and the hand-crafted feature could further increase the performance. It demonstrates that the effect of RASF is somehow orthogonal to these hand-crafted features. RASF brings additional notable improvements over the existing methods.

## 6 CONCLUSION

We propose Representation-Agnostic Shape Fields (RASF), a generalizable and computation-efficient shape embedding layer for 3D deep learning. Shape embeddings for various 3D shape representations (point clouds, meshes and voxels) are retrieved by coordinates indexing. We provide two effective schemes for RASF pre-training, that is shape reconstruction and normal estimation, to enable RASF to learn robust and general shape embeddings. Once trained, RASF could be plugged into any 3D neural network with negligible cost. RASF widely boosts the performance for various 3D representations, neural backbones and applications.

## 7 ACKNOWLEDGEMENT

This work was supported by National Science Foundation of China (U20B2072, 61976137).

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

## A  APPENDIX

### A.1  DETAILS OF PRE-TRAINING RASF

To demonstrate the general effectiveness of RASF, we use the same RASF weights in all the experiments below. We pre-train RASF based on a large point clouds dataset, ShapeNetPart (Yi et al., 2016). It includes training samples of $12,137$ and testing samples of $2,847$. We randomly sample $2,048$ points for each object as the input shape to RASF. We set the resolution $R$ of RASF to $16$ and the channel dimension $C$ to $32$. During the pretraining in ShapeNetPart, we set the number of neighbors $K$ to $64$ for 2048 points per sample. $K$ changes adaptively according to the total number of points in one sample, so as to preserve a relatively stable scale in RASF across different shapes. In the reconstruction pre-text task, we sample $N_s = 24$ rows in the $2048 \times 32$ shape embeddings to feed into the reconstruction network. The choosing of the RASF hyper-parameters are analyzed in Sec. A.6.

The training lasts for 150 epochs, with an initial learning rate of $0.001$ using Adam. We decay the learning rate by $0.2$ for every 50 epochs. The chamfer distance on the test-set converges to $0.003$ at the end. It is hard to tell the difference between the reconstructed shape and the ground-truth with human eyes.

### A.2  DETAILS OF IMPLEMENTING RASF

When RASF is used in down-stream tasks, it is used only before the first layer in the backbones networks, making it very simple to implement. Suppose RASF embeddings have $C$ channels and the original backbone receives input of $C_o$ channels. We simply enlarge the channels of the first layer to $C + C_o$ to implement backbones with RASF. All other settings are identical between bakcbones with and without RASF.

### A.3  VISUALIZATION OF RECONSTRUCTED SHAPE

We illustrate the reconstructed shapes during the reconstruction task for RASF pre-training in Fig. 4. The predicted shapes are obtained on test-set after convergence. As shown in the figure, the reconstructed shapes and the ground truths are hardly distinguishable with human eyes, demonstrating that RASF has learned the local geometry for favourable reconstruction performance.

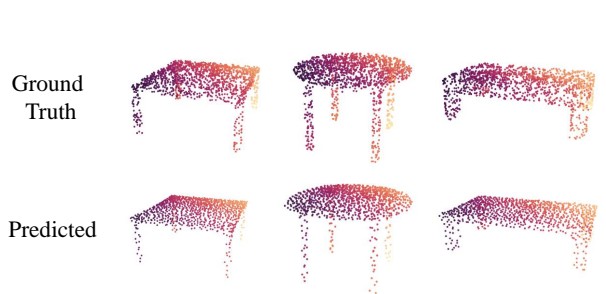
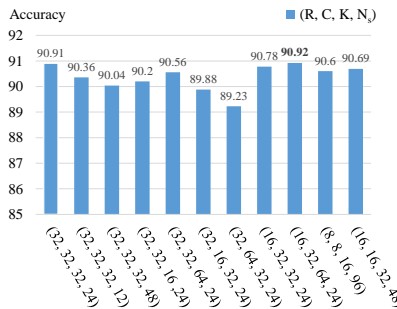

Figure 4: The predicted shapes and ground-truths during the reconstruction task.

Figure 5: Sensitivity analysis of RASF hyper-parameters.

### A.4  VISUALIZATION OF RASF

We design two approaches to visualize the pre-trained RASF. One is directly illustrating the weights. The other is feeding RASF with a set of semi-ellipsoids with different curvatures to explore how RASF would respond to geometrically-varying shapes.

**Illustrations of RASF Weights.**  We visualize each channel of the pre-trained RASF. First, we visualize the three-dimensional grid using *visvis*, a python library (Klein, 2020). Besides, we reduce the three-dimensional grid of each channel to two-dimensional by conducting max-pooling on the

x-axis (the other two axes have a similar phenomenon). The max-pooling here corresponds to the max-pooling in RASF to obtain shape embeddings. As illustrated in Fig. 6, we find that each channel focus on a different area, representing a particular local shape. Besides, some of the channels show a characteristic of symmetry, which might represent a symmetrical local shape.

**Visualization with Deformed Ellipsoids.** We explore how RASF would respond to geometrically-varying shapes by designing a proof-of-concept experiment. We generate a set of semi-ellipsoids in diverse curvatures by varying the radius lengths on the three axes. A portion of the generated semi-ellipsoids are illustrated in Fig. 7 (left). A semi-ellipsoid could be parameterized by spherical coordinates. Suppose the semi-ellipsoid axes coincide with coordinate axes, we have:

$$x = asin(\theta)cos(\varphi) \tag{2}$$
$$y = bsin(\theta)sin(\varphi) \tag{3}$$
$$z = ccos(\theta) \tag{4}$$

where $a, b, c$ denote the radius lengths on x-axis, y-axis and z-axis, $0 \leq \theta \leq \pi/2, 0 \leq \varphi \leq 2\pi$. To vary the radius length along the x-axis, we set $b, c$ as 1 and increases a from 0.1 to 2 in a step of 0.1, so as y-axis and z-axis. In this way, we generate three groups of semi-ellipsoids, each of which contains 20 semi-ellipsoids in different shapes for one axis. We feed these shapes into the pre-trained RASF and obtain their shape embeddings. We consider the peak of the semi-ellipsoids as the central point of RASF, which are marked red in Fig. 7 (left), while the others are considered as the local shape of the peak. At last, we obtain 20 shape embedding vectors for 20 shapes in each group. We illustrate the shape embeddings by arranging them in order in rows, as shown in Fig. 7 (right).

It is observed that some channels (channel 6, 11, 12, 20, 25, 29, 30) have strong correspondence to the ellipsoid curvature. The response of these channels changes gradually from large curvature shapes to small curvature shapes. On the other hand, some channels have the same response given different curvature shapes. We suppose these channels could be related to other geometries, such as cones or cubes.

## A.5 DATASETS IN DOWNSTREAM TASKS

**ModelNet.** The ModelNet datasets are introduced by Wu et al. (Wu et al., 2015) for 3D object classification. ModelNet40 includes 40 categories of 3D rigid objects. ModelNet10 is a subset of ModelNet40. Point clouds (Qi et al., 2017a) and volumetric representations (Wu et al., 2015) are available for this dataset.

**ShapeNetPart.** ShapeNetPart (Yi et al., 2016) is a point cloud dataset for benchmarking 3D shape segmentation. It contains $16, 881$ shapes from 16 categories. Each point in the point cloud is annotated with one of 50 labels. Most categories are labeled with two to five parts. The size of the train-set and test-set are $12, 137$ and $2, 847$ respectively.

**S3DIS.** The Stanford Large-Scale 3D Indoor Spaces (S3DIS) (Armeni et al., 2016) is for a semantic indoor scene segmentation task, containing 6 indoor areas with 271 rooms. Each point is annotated with one of the 13 categories, e.g., board, chair, ceiling, etc. plus clutter. We conduct cross-validation on the 6 areas, the same protocol as prior works (Armeni et al., 2016; Qi et al., 2017a; Wang et al., 2019).

**SHREC.** SHREC (Lian et al., 2011) is a 30-classes dataset for mesh classification, with 20 examples for each class. The categories include rigid objects such as lamps, and also non-rigid objects such as aliens. We follow the protocol in (Ezuz et al., 2017), which generates two kinds of split for training and testing. The first is to randomly sample 10 examples from 20 per class for training to form *SHREC10*, yielding a 1 : 1 train-test-split. The other is to randomly sample 16 from 20 for training, named *SHREC16*. Since (Ezuz et al., 2017) did not release their train test split, we conduct the random split ourselves.

**HUMAN.** The HUMAN dataset is proposed by Maron et al. (Maron et al., 2017) for mesh segmentation. The train-set includes 370 models from SCAPE (Anguelov et al., 2005), FAUST (Bogo et al., 2014), MIT (Vlasic et al., 2008) and Adobe Fuse (Adobe, 2016), while the test-set includes 18 models from SHREC07 (Giorgi et al., 2007) (humans) dataset. The 388 models in total are manually annotated with 8 labels based on (Kalogerakis et al., 2010).

A.6    HYPER-PARAMETERS.

We analyze the sensitivity of the hyper-parameters in RASF. There are four hyper-parameters in RASF, that is the resolution $R$, the channel $C$, the number of neighbor points $K$ to obtain shape embedding, and the number of shape embeddings $N_s$ to feed into the reconstruction network. We change one parameter at a time, using the set of hyper-parameters to pre-train the RASF and adopt it in the downstream task. At last, we decrease the $R$, $C$ and the neighbor points, while increasing $N_s$ at the same time, given that the reconstruction network needs more input shape embeddings when RASF models smaller shapes. We adopt PointNet (Qi et al., 2017a) on ModelNet40 (Wu et al., 2015) for demonstration. The results are shown in Fig. 5. The results show that RASF grid needs a proper size to achieve the best performance. However, increasing the size (resolution and channel) brings more harm than decreasing it. We argue that RASF needs a proper resolution and number of channels to achieve optimal performance.

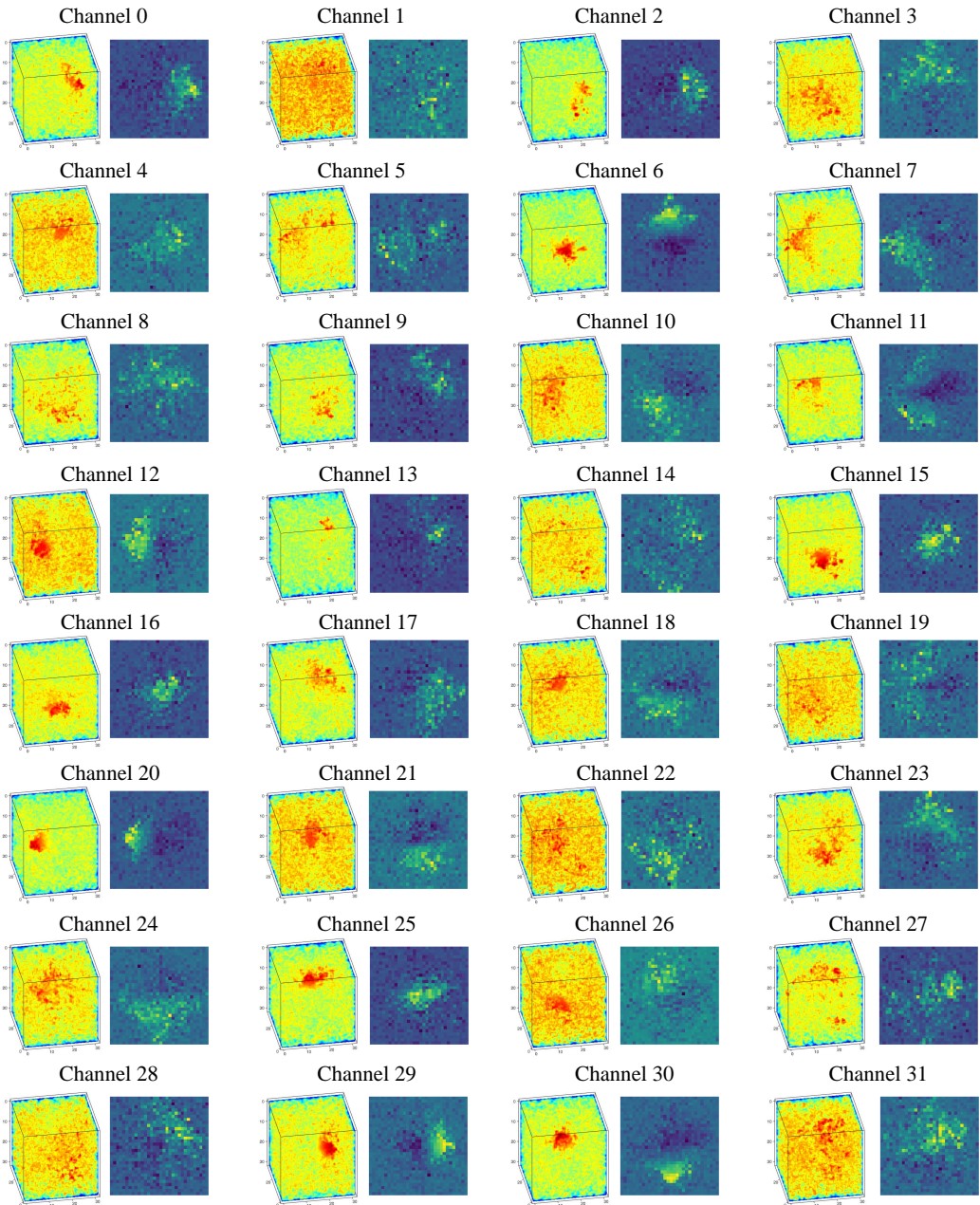

Figure 6: Visualization of RASF weights. For each channel, we show the 3D visualization and the 2D figure after max-pooling on the x-axis. It is observed that each channel focus on a different area, representing a particular local shape.

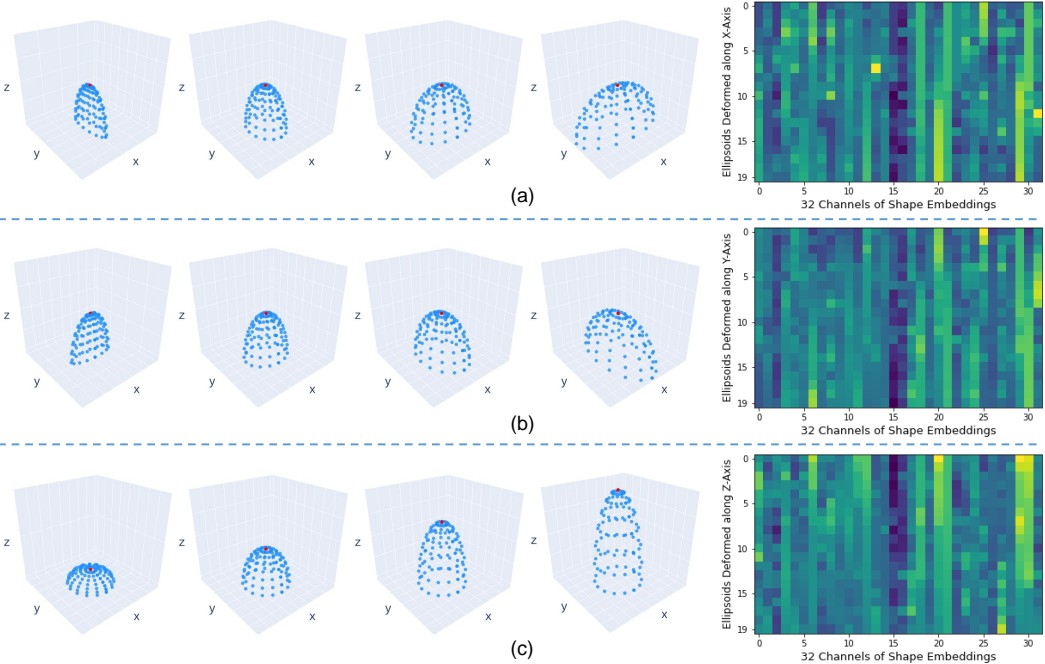

Figure 7: *left*: The input semi-ellipsoids with different curvatures. *right*: The visualization of shape embeddings given three groups of semi-ellipsoids input. (a)(b)(c) are the three groups of semi-ellipsoids (deformed along three axes) with their corresponding shape embeddings. The shape embeddings are arranged in order row by row. Each row represents one shape embedding. It is observed that the response of several channels changes gradually from large curvature shapes to small curvature shapes.

