# OpenReview forum: "Representation-Agnostic Shape Fields"
_ICLR.cc/2022/Conference — ICLR 2022 Poster_

### Official Review · Reviewer_uSYH · 2021-10-23

**Correctness:** 3
**Technical Novelty And Significance:** 2
**Empirical Novelty And Significance:** 2
**Recommendation:** 5
**Confidence:** 4

**Details Of Ethics Concerns:**

Nil.

**Main Review:**

Strengths:

(1) This work has a good motivation of trying to unify various 3D representations: meshes, point clouds, and voxels.

(2) The design is simple and brings certain improvements to most (basic) networks employed in the experiments, and the method can be used a plugin for various method

(3) The paper is well written and quite clear


Weaknesses:

(1) The first concern is on the motivation of the design. Though the paper claims "generalizable", the method simply re-samples points for mesh and voxel inputs, so that we can obtain points for any kind of inputs and then use these points for local shape extraction and embedding.  So, it seems to me that it is too strong to claim that the method is generalizable.

(2) From the results shown in Section 4, the method seems to be effective mainly for earlier networks such as PointNet (2018) and MeshCNN (2019), which explore very local information in the input.

(3) The idea of finding K nearest neighbors for feature extraction sounds very similar to EdgeConv in DGCNN, which improves over PointNet with features from local neighborhood of K nearest points. From Table 3, the improvement of the proposed method over DGCNN is quite marginal.

(4) I am also concerned about the effectiveness of the design.  As shown in Table 8, the randomly-initialized RASF achieves comparable performance with the pre-trained one. If the backbone is changed from PointNet to DGCNN, it may be hard to see such improvement.

**Summary Of The Paper:**

This paper presents a method for 3D shape analysis by means of local shape embedding.  The key idea is to use a learnable multi-channel 3D grid to embed local shapes in the input 3D object, similar to word embedding in NLP; hence, the method can be pre-trained and applied for various downstream tasks.

The procedure is:
(1) If the input is mesh-based or voxel-based, first generate sample points on mesh or in voxels; if input is point-based, no need to have this pre-processing;
(2) For each point p, find a set of K nearest neighbors points, i.e., P_neigh; and
(3) normalize { p, P_neigh } and take the result as an index to retrieve shape features from the 3D grid, etc.

The paper claims that the proposed method is generalizable (i.e., could be used in different 3D representations, backbones and downstream tasks) and computation-efficient shape embedding layer for 3D deep learning.

**Summary Of The Review:**

Overall, I am lukewarm for this paper and slightly more on the negative side.


--------------------------------------


Other issues:

I am not sure how sensitive the method is to K and the density of the point samples in the inputs.


P.3:

descriptors( -> descriptors (

(Sun et al., 2009) , -> (Sun et al., 2009),


P.6:

a accuracy -> an accuracy

ShapenetPart -> ShapeNetPart

representation, We -> representation, we

classfication -> classification


P.7:

Adam(Kingma -> Adam (Kingma

---

> ### Author Response · Authors · 2021-11-21
> **Response to Reviewer uSYH (Part 1/2)**
>
> We thank you for your detailed comments. We address your questions as follows:
>
> **Q1**:
> > The first concern is on the motivation of the design. Though the paper claims "generalizable", the method simply re-samples points for mesh and voxel inputs, so that we can obtain points for any kind of inputs and then use these points for local shape extraction and embedding. So, it seems to me that it is too strong to claim that the method is generalizable.
>
> **A1**: Please note that there are two kinds of “point” in our study: 1) “point cloud” as a data structure in 3D geometry, and 2) “point-based graphics” [1] as a way to solve problem in geometric processing. We use a “point-based” approach to improve the performance of 3D deep learning, which is “generalizable” across various shape representation including voxel, mesh and “point cloud”. The term “generalizable” refers to: 1) The obtained RASF embeddings are used as additional features for the original data representations, so that the data could still be processed by their corresponding backbones, such as MeshCNN for meshes, VoxNet for voxels. We do not transform meshes or voxel into point clouds to feed into a point-cloud-based backbone. Each shape representation has unique advantages on specific scenarios, while the flexibility of the RASF makes it easy to be integrated with any representation. 2) The RASF is proven to be effective for different representations. It is pre-trained on ShapeNetPart, a shape dataset based on point clouds while used for various 3D representations, tasks and datasets, not only point clouds.
>
> [1] Gross, Markus, and Hanspeter Pfister, eds. Point-based graphics. Elsevier, 2011.
>
> **Q2**:
> > From the results shown in Section 4, the method seems to be effective mainly for earlier networks such as PointNet (2018) and MeshCNN (2019), which explore very local information in the input.
>
> **A2**: We are not sure of the meaning of the term “local” here. To our understand, PointNet and MeshCNN have very different mechanisms. PointNet aggregates the point feature by a single global pooling layer, while MeshCNN aggregates neighbor feature by repeatedly pooling in local space. PointNet should be more global and MeshCNN should be more local. RASF is effective in both mechanisms. Besides, based on Reviewer ESBS’s suggestion, we additionally experiment on a recent state-of-the- art point cloud backbone, PCT [1], which learns both local and global geometry. We run PCT without RASF and with RASF under the same setting, and the ACCs are 92.93% and 93.18%, respectively, where RASF improves over PCT by 0.25%. It demonstrates that RASF could consistently make improvements over diverse backbones, no matter it is more local or global. We highlight the added value of our method lies in the generalizability, usability and low cost regardless of shape representation, tasks, datasets and network backbones for 3D deep learning.
>
> **Q3**:
> > The idea of finding K nearest neighbors for feature extraction sounds very similar to EdgeConv in DGCNN, which improves over PointNet with features from local neighborhood of K nearest points. From Table 3, the improvement of the proposed method over DGCNN is quite marginal.
>
> **A3**: We agree that both EdgeConv in DGCNN and RASF make use of finding K nearest neighbors. However, KNN is indeed a common method in 3D shape analysis, which is used in many 3D models. Beyond the KNN as an implementation detail in our framework, the novelty of the proposed RASF is representing local 3D geometry by discrete 3D grids; therefore, the semantic feature for various shape representations can be extracted from this simple design. What’s more, we have empirically proved that EdgeConv does **NOT** generalize to meshes in Section 5.1 (Table 7), where the RASF does.
>
> **Q4**:
> > I am also concerned about the effectiveness of the design. As shown in Table 8, the randomly-initialized RASF achieves comparable performance with the pre-trained one. If the backbone is changed from PointNet to DGCNN, it may be hard to see such improvement.
>
> **A4**: Thanks for your comment. To address your concern, we additionally experiment on the DGCNN with a randomly-initialized RASF. We list the results below:
>
> | Setting                                   | ACC on ModelNet40 |
> |----                                       |       ----        |
> |DGCNN without RASF                         |92.71              |
> |DGCNN with pre-trained RASF                |92.95              |
> |DGCNN with randomly-initialized RASF       |92.52              |
>
> It shows that randomly-initialized RASF still degrades the performance, even if the backbone is DGCNN. This result is consistent with the results in Table 8, where the backbone is PointNet. We reckon that a randomly-initialized RASF that is jointly trained with the backbone would overfit to the training set, while a pre-trained RASF is more generalizable.

---

> > ### Author Response · Authors · 2021-11-21
> > **Response to Reviewer uSYH (Part 2/2)**
> >
> >
> > **Q5**:
> > >  I am not sure how sensitive the method is to K and the density of the point samples in the inputs.
> >
> > **A5**: The number K in KNN and the density of point samples could both be converted to the number of the neighbor points, since we linearly scale up K as a function of the total point number in our experiments, such as K=32 when the sample has 1024 point and K=64 when the sample has 2048 points. We analyzed the sensitivity of K in Section A.5 in the appendix (the 1st, 4th and 5th columns in Figure 5). We tried different numbers of K (16, 32, 64), and find that they yield results between 90.2%-90.9%.
> >
> > **A6**: Thanks for pointing out the typos. We have fixed them.

---

### Official Review · Reviewer_quxB · 2021-10-29

**Correctness:** 4
**Technical Novelty And Significance:** 3
**Empirical Novelty And Significance:** 3
**Recommendation:** 6
**Confidence:** 4

**Details Of Ethics Concerns:**

No.

**Main Review:**

### Strength
- The proposed shape embedding is general and can be plug-and-play in various 3D representations.
- The proposed method has been extensively evaluated with different representations in a range of downstream applications, including classification, part segmentation, and semantic segmentation. All the experiments have witnessed a boost in performance when incorporating the proposed shape embedding, which indicates the effectiveness of the proposed approach.
- The paper is well written and easy to follow.

### Weakness
- Though the performance can be boosted using the proposed method, the increase of the quantitative measurement seems to be a bit marginal.

**Summary Of The Paper:**

This paper presents a generalizable shape embedding layer for 3D deep learning. The proposed shape embedding can be used for a number of 3D shape representations including point cloud, mesh, and voxel. The shape embeddings can be obtained via self-supervised learning, where the paper has experimented with shape reconstruction and normal estimation as the pre-training tasks. The pre-training would enable the proposed method to learn general shape embeddings. At deploy time, the 3D priors are retrieved using coordinate indexing. The experimental results have shown that the proposed method can provide a boost on various representations in different applications.

**Summary Of The Review:**

The idea of introducing an embedding layer to the 3D shape analysis just as the NLP community is interesting and novel.
The proposed framework is general to many mainstream 3D representations and does not require many additional changes to existing 3D learning frameworks in order to be deployed. The experimental results are positive and have shown the effectiveness of the proposed method in a number of applications. Though the performance boost has been verified, the magnitude of the increase seems to be a bit marginal.

---

> ### Author Response · Authors · 2021-11-21
> **Response to Reviewer quxB**
>
> We thank you for your comments and positive feedback. We address the comment as follows:
>
> **Q1**:
> > Though the performance can be boosted using the proposed method, the increase of the quantitative measurement seems to be a bit marginal.
>
> **A1**: We highlight the added value of our method lies in the generalizability, usability and low cost regardless of shape representation, tasks, datasets and network backbones for 3D deep learning. Besides, the improvements in some experiments are notable, such as mesh classification and segmentation. We believe this “simple yet effective” idea can be an addition to geometric deep leaning community.

---

### Official Review · Reviewer_ESBS · 2021-11-01

**Correctness:** 3
**Technical Novelty And Significance:** 3
**Empirical Novelty And Significance:** 3
**Recommendation:** 5
**Confidence:** 4

**Main Review:**

Overall I found the idea behind RASF compelling. The concept is simple and yet it does boost accuracy while keeping the added computational cost to a reasonable degree.

Strengths:
- Clear idea and exposition. I like that the authors didn't try to over-complicate the exposition of the idea.

- Good evaluation on multiple geometry representations for different problems (reconstruction, normal estimation, segmentation, classification). It seems that adding RASF improves the accuracy for all the methods.


Weaknesses:
- Implementation details are unclear sometimes. Are RASF embedding layers used for all layers in pointnet for classification? I think the reader would benefit from an appendix where implementation details are clear.

- The baselines used to add RASF on top might be outdated. It would be good if the authors could provide at least a couple of comparisons where baselines are state-of-the-art. Specially, those architectures using transformers (eg. https://arxiv.org/abs/2012.09688), which can learn both local and global geometry. I would like to see these comparisons before recommending this paper to be accepted.


Finally, I have a couple of curiosities that I would like the authors to provide some feedback if possible:

- Have you though about using RASF at different scales? One could easily do this by having multiple RASF embeddings which are queried by neigborhoods at different scales?

- What would be the expected performance of a transfer learning task? Where you train RASF on point clouds and then test it on meshes? This could be an interesting direction for future work,

**Summary Of The Paper:**

This submissions introduces RASF (Representation Agnostic Shape Fields), an embedding layer that encodes local geometry and can be used for 3D deep learning with different input domains: point clouds, meshes or voxels. RASF is inspired by the idea of embedding layers in language models, where representations of tokens are indexed by one-hot vectors.

In this submissions the authors propose to implement RASF as volumetric latent embedding that is indexed via tri-linear sampling. For example, given an input pointcloud the process first extracts a local neighborhood around a point p and normalizes it, so that the point p becomes the origin. This pointcloud is then used to sample the volumetric latent embedding and afterwards a max-pooling operation is applied. The max-pooled feature becomes the representation of point p.

The experimental results show improvements when RASF is used on top of several baselines while incurring on a small additional computational complexity.

**Summary Of The Review:**

This paper presents a conceptually simple approach that seems to boost performance for most baseline models for 3D deep learning. I like the exposition of the idea and the extensive experimental results. However, I'm concerned that the baselines chosen for comparison might be outdated. If the authors can provide results with updated baselines (specially those that use transformers as backbones) I would be happy to upgrade my current score.

---

> ### Author Response · Authors · 2021-11-21
> **Response to Reviewer ESBS**
>
>
> We thank you for your detailed and constructive comments. We address each of your concerns as follows:
>
> **Q1**:
> > Implementation details are unclear sometimes. Are RASF embedding layers used for all layers in pointnet for classification? I think the reader would benefit from an appendix where implementation details are clear.
>
> **A1**: The RASF embedding layers are only used as additional features, before the first layer in all backbones, which is straightforward to implement. We have added more implementation details in the appendix according to your suggestions. Besides, the code will be open-source upon acceptance.
>
> **Q2**:
> > The baselines used to add RASF on top might be outdated. It would be good if the authors could provide at least a couple of comparisons where baselines are state-of-the-art. Specially, those architectures using transformers (eg. https://arxiv.org/abs/2012.09688), which can learn both local and global geometry. I would like to see these comparisons before recommending this paper to be accepted.
>
> **A2**: Thanks for your comments. According to your suggestion, we have added PCT in the experiment. Due to the limit of time, so far we only have results on the classification task on ModelNet40. Based on the official code (https://github.com/Strawberry-Eat-Mango/PCT_Pytorch), we run the PCT without RASF and PCT with RASF under the same setting. The ACCs are 92.93% and 93.18% respectively, where RASF improves over PCT by 0.25%. We have included these results and cited the reference in the revised manuscript.
>
> **Q3**:
> > Have you thought about using RASF at different scales? One could easily do this by having multiple RASF embeddings which are queried by neighborhoods at different scales?
>
> **A3**: We have implemented a naïve multi-scale extension of RASF  by concatenating multiple RASF embeddings queried by neighborhoods at different scales, in three settings, as listed in the table:
>
> | Number of neighbors in RASF concatenation | ACC on ModelNet40 |
> |----                                       |       ----        |
> |16,32                                      |88.73              |
> |16,32,64                                   |87.07              |
> |32,64                                      |90.84              |
>
> It implies that with the existence of 16-neighbors RASF, the performance would degrades obviously. It could be explained by that the scale of neighbors is fixed during the pre-training of RASF; therefore, it could not adapt to various scales in down-stream tasks. Nevertheless, the multi-scale idea could further improve the performance with sophisticated implementation. In future work, we will try different neighbor scales in multiple RASF pre-training and use them together in down-stream tasks. Besides, adaptive scales are also in our considerations.
>
> **Q4**:
> > What would be the expected performance of a transfer learning task? Where you train RASF on point clouds and then test it on meshes? This could be an interesting direction for future work.
>
> **A4**: There might be a misunderstanding. RASF is pre-trained on ShapeNetPart, a point clouds dataset, as stated in the last paragraph in Section 3.3. Once RASF is pre-trained, it is fixed in **ALL** down-stream tasks, including point clouds, meshes and voxel grids. Therefore, we claim that RASF is an off-the-shelf module to improve performance at negligible cost in adaptation of down-stream tasks.

---

### Official Review · Reviewer_Y3fs · 2021-11-03

**Correctness:** 4
**Technical Novelty And Significance:** 3
**Empirical Novelty And Significance:** 4
**Recommendation:** 6
**Confidence:** 4

**Main Review:**

This paper takes inspiration from word embeddings in NLP and applies the idea in a novel way to 3D deep learning. It is a simple idea, and it is exciting that it appears to be effective across various tasks and modalities. I think moving towards "universal" shape representations is an interesting direction for 3D deep learning, and this paper makes a nice step in that vein.

It is shown in the appendix that increasing the resolution and number channels in the embedding grid degrades quality, which is surprising. The authors claim that this is due to overfitting, but is this quantitatively validated? It would be nice to see metrics on the training set if this is indeed the case.

The authors should explicitly confirm that the learning set-ups are identical for each backbone with and without RASF---i.e., that the test/train splits are the same and architectures only differ in the embedding layer. It also might be helpful to include some error bars across different random seeds.

Have you tried using your approach on some more recent state-of-the-art 3D learning methods? The ones compared against are fairly representative but a bit outdated, so I wonder if latest advances might make this method obsolete.

While the paper is generally clear, there are typos and minor issues in exposition, some of which are listed below. In particular, I would encourage the authors to make some adjustments to some of the figures so that they are easier to parse just from the illustrations and caption contents.


Figure 1 is difficult to parse. Some description of the actual contents of the plot in the caption would make it more clear.

The introduction has very many citations that are not particularly central to the paper and are mostly cited again in the related works section. It might be worth removing some of them from the introduction, because currently it makes the text appear somewhat cluttered and difficult to read.

Page 1: "In recent computer vision and deep learning community..." sentence needs some restructuring

Page 1: "coordinate lacks geometric information" not entirely clear what this means at this point in the paper

Page 2: "indices by" -> "indexed by"

DGCNN is mentioned several times but with the wrong citation---the actual paper is [Wang et al. 2018].

"HodgeNet: Learning Spectral Geometry on Triangle Meshes" [Smirnov and Solomon 2021] should be mentioned in the related work on learning on meshes.

The description of the point extraction and normalization in 3.1 is a bit hard to follow.

Page 5: "by visualize" -> "by visualizing"

Figure 3 is very small, and it is not immediately clear what it's trying to show. It would help to add some more labeling.

Figure 3 caption: "geometrically-various" -> "geometrically-varying"

**Summary Of The Paper:**

This paper proposes a method for learning a latent spatial embedding of points in space to a feature space by pre-training on a pretext task (e.g., reconstruction). This embedding can the be used as part of any deep learning pipeline that inputs point coordinates as part of its architecture by first mapping the coordinates to their embedding. The authors demonstrate that utilizing these embeddings improves results of state-of-the-art learning methods on point clouds, meshes, and voxel grids.

**Summary Of The Review:**

I think this paper proposes a simple yet effective framework for pertaining embedding for 3D deep learning tasks. Despite some minor questions and issues in exposition, I think it would be a nice addition to ICLR.

---

> ### Author Response · Authors · 2021-11-21
> **Response to Reviewer Y3fs**
>
>
> We thank you for your detailed comments and positive rating for the added value of our method in 3D deep learning. We address your questions as follows:
>
> **Q1**:
> > It is shown in the appendix that increasing the resolution and number channels in the embedding grid degrades quality, which is surprising. The authors claim that this is due to overfitting, but is this quantitatively validated? It would be nice to see metrics on the training set if this is indeed the case.
>
> **A1**: The term “overfitting” in the pre-resived manuscript might not be precise, since the setting of our experiments is not a standard train-test setting: we first pre-train the RASF on a large-scale shape dataset (ShapeNetPart), and use it on various downstream datasets where the RASF is fixed and other models with the RASF are trained in a standard train-test setting. We have conducted experiments where we pre-train and use RASF on the same dataset (ModelNet40), and find that increasing the resolution and number channels still degrades the performance. Therefore, wWe have changed the “overfitting” explanation in the manuscript into a more precise statement: The RASF needs a proper number of channels and resolution to achieve optimal performance.
>
> **Q2**:
> >The authors should explicitly confirm that the learning set-ups are identical for each backbone with and without RASF---i.e., that the test/train splits are the same and architectures only differ in the embedding layer. It also might be helpful to include some error bars across different random seeds.
>
> **A2**: We confirm that all settings are identical for each backbone with and without the RASF, including hyper-parameters and train-test splits. We have clarified this in the revised manuscript. The error bars have not been included in the current version due to the limit of computation resources. We will add it in the final manuscript by repeated experiments.
>
> **Q3**:
> >Have you tried using your approach on some more recent state-of-the-art 3D learning methods? The ones compared against are fairly representative but a bit outdated, so I wonder if latest advances might make this method obsolete.
>
> **A3**: Thanks for your suggestion. As suggested by other reviewer (Reviewer ESBS), we experiment on a more recent point cloud backbone based on transformer, PCT [1], which achieves SOTA performance in many tasks. We run PCT without RASF and with RASF under the same setting. The ACCs are 92.93% and 93.18% respectively, where RASF improves over PCT by 0.25%. We have included these results in the revised manuscript.
>
> **Q4**:
> >Figure 1 is difficult to parse. Some description of the actual contents of the plot in the caption would make it more clear. Figure 3 is very small, and it is not immediately clear what it's trying to show. It would help to add some more labeling.
>
> **A4**: We have improved the clarity of captions and sizes of Figure 1 and Figure 3 for better presentation.
>
> **A5**: All typos and the mistaken reference have been fixed in the revised manuscript. The redundant references in introduction have been removed. HodgeNet [2] has been added to the related work.
>
>
> [1] Guo, Meng-Hao, et al. "PCT: Point cloud transformer." Computational Visual Media 7.2 (2021): 187-199.
>
> [2] Smirnov, Dmitriy, and Justin Solomon. "HodgeNet: Learning Spectral Geometry on Triangle Meshes." SIGGRAPH (2021).

---

### Decision · Program_Chairs · 2022-01-20

**Decision:**

Accept (Poster)

**Comment:**

This paper proposes an "embedding layer" in which points on a model are mapped into a feature space, trained using a reconstruction-based pretext task.  Then, the resulting embedding layer can be applied to shape data before using different learning architectures for modalities like meshes and point clouds.  The work is particularly interesting in its attempt to derive a learned shape representation that is agnostic to modality.  Some questions remained about experiments (e.g. baselines), but these are relatively minor and partially addressed in the rebuttal phase; also, sometimes the improvement seems to be marginal in practice.

Two reviewers championed this work during the discussion phase.  The AC tends to agree this work is an interesting direction for future work and contains insight that the vision/learning communities might be able to use in other settings.